# Identification of MGMT Downregulation Induced by miRNA in Glioblastoma and Possible Effect on Temozolomide Sensitivity

**DOI:** 10.3390/jcm12052061

**Published:** 2023-03-06

**Authors:** Andrea Cardia, Samantha Epistolio, Ismail Zaed, Nora Sahnane, Roberta Cerutti, Debora Cipriani, Jessica Barizzi, Paolo Spina, Federico Mattia Stefanini, Michele Cerati, Sergio Balbi, Luca Mazzucchelli, Fausto Sessa, Gianfranco Angelo Pesce, Michael Reinert, Milo Frattini, Francesco Marchi

**Affiliations:** 1Service of Neurosurgery, Neurocenter of the Southern Switzerland, Regional Hospital of Lugano, Ente Ospedaliero Cantonale (EOC), 6900 Lugano, Switzerland; 2Laboratory of Molecular Pathology, Institute of Pathology, Ente Ospedaliero Cantonale (EOC), 6900 Locarno, Switzerland; 3Unit of Pathology, Department of Medicine and Surgery, University of Insubria, ASST Sette Laghi, 21100 Varese, Italy; 4Department of Enviromental Science and Policy, Faculty of Science and Technology-ESP, University of Milan, 20122 Milan, Italy; 5Division of Neurological Surgery, Department of Biotechnology and Life Sciences, University of Insubria, ASST Sette Laghi, 21100 Varese, Italy; 6Radiation Oncology, Oncology Institute of Southern Switzerland, Ente Ospedaliero Cantonale (EOC), 6501 Bellinzona, Switzerland; 7Faculty of Medicine, University of the Southern Switzerland, 6900 Lugano, Switzerland

**Keywords:** glioblastoma, MGMT, miRNA, overall survival, progression-free survival, temozolomide

## Abstract

Glioblastoma multiforme (GBM) remains one of the tumors with the worst prognosis. In recent years, a better overall survival (OS) has been described in cases subjected to Gross Total Resection (GTR) that were presenting hypermethylation of Methylguanine-DNA methyltransferase (MGMT) promoter. Recently, also the expression of specific miRNAs involved in MGMT silencing has been related to survival. In this study, we evaluate MGMT expression by immunohistochemistry (IHC), MGMT promoter methylation and miRNA expression in 112 GBMs and correlate the data to patients’ clinical outcomes. Statistical analyses demonstrate a significant association between positive MGMT IHC and the expression of miR-181c, miR-195, miR-648 and miR-767.3p between unmethylated cases and the low expression of miR-181d and miR-648 and between methylated cases and the low expression of miR-196b. Addressing the concerns of clinical associations, a better OS has been described in presence of negative MGMT IHC, in methylated patients and in the cases with miR-21, miR-196b overexpression or miR-767.3 downregulation. In addition, a better progression-free survival (PFS) is associated with MGMT methylation and GTR but not with MGMT IHC and miRNA expression. In conclusion, our data reinforce the clinical relevance of miRNA expression as an additional marker to predict efficacy of chemoradiation in GBM.

## 1. Introduction

Among the intracranial pathologies, glioblastoma multiforme (GBM) remains the most common malignant primary tumor in adult patients. Despite the decades of research and technological improvements, it is still one of the tumors with the worst prognosis, with a median overall survival (OS) of 12–15 months from time of diagnosis [1,2]. The gold standard management remains safe optimal surgical resection followed by adjuvant partial brain radiotherapy combined with concomitant and adjuvant chemotherapy with temozolomide (TMZ) [3]. Recent studies demonstrated significant correlation between extent of resection (EOR) and OS in patients with GBM, especially in case of Gross Total Resection (GTR) compared to Subtotal Resection (STR) [3,4,5].

In more recent years, scientific research focused on the role of Methylguanine-DNA methyltransferase (MGMT) in the management of GBM. It has been noticed that hypermethylation of MGMT promoter (a post-transcriptional mechanism leading to absence of MGMT protein expression) leads to improved response to TMZ, thus improving the patients’ outcome [3,6,7,8,9,10].

It has been also observed that there are other factors affecting MGMT expression in GBM. Some studies have shown that some patients with unmethylated tumors may experience an unexpected favorable outcome after radio-chemotherapy; in these cases, mRNA expression was found to be low [11]. One of these mechanisms affecting mRNA expression seems to be represented by microRNA (miRNA) expression. There are now data suggesting how changes in miRNA expression may lead to the degradation of MGMT mRNA, with the result of MGMT gene silencing [11,12]. Indeed, these miRNAs affect GBM phenotype transition and malignant progression targeting more than 500 targets responsible for various biological processes such as cell proliferation, division, growth, and intercellular communication [11,12,13].

In the study here presented, the relevance of the data regarding the influence, in GBM, of specific miRNAs (i.e., miR-21, miR-195, miR-767-3p, miR-196b, miR-648, miR-181d, miR-181c) [14,15,16] on MGMT and consequently on prognosis, led us to investigate the pattern of miRNA expression and its correlation to TMZ sensitivity by analyzing a large cohort of GBM patients. The molecular data have been then correlated with patients’ clinical outcome.

The primary objective of the study is to verify whether a pattern of miRNA expression correlates with response to the treatment with TMZ and its clinical efficacy, while the secondary objective is to verify if a cumulative panel of markers may identify a group of patients experiencing a better outcome after the administration of standard TMZ therapy combined with radiotherapy.

## 2. Materials and Methods

In order to fulfil the objectives of the paper, data were collected retrospectively from 2 tertiary neurosurgical centers of different European countries (Service of Neurosurgery of the Neurocenter of Southern Switzerland, EOC, Switzerland and Department of Neurosurgery at Insubria University Hospital, Italy) over a ten-year period (2004–2013). The study was conducted in compliance with protocol, the current version of the Declaration of Helsinki, the ICH-GCP or ISO EN 14155 [17] (as far as applicable) as well as all national legal and regulatory requirements. Data and samples have been collected and analyzed for the study purpose only after the required authorizations from the competent Ethics Committees (Cantonal Ethics Committee, Bellinzona, Switzerland) were obtained (Rif. CE 3086-2016-01108).

For each patient, data include gender, age, type of surgery, postoperative outcome, postoperative complications and general follow-up until the death of the patients. All the data are obtained from the electronic medical registries of the 2 institutions.

Inclusion criteria were age >18 years, histological diagnosis of IDH1 wt WHO grade IV, therapy with TMZ according with the Stupp scheme (60 Gray radiotherapy and concomitant chemotherapy with TMZ, followed by six cycles of maintenance TMZ), death caused by GBM, tissue availability for biomolecular analyses.

The exclusion criteria were represented by no clear diagnosis of GBM or presence of low-grade gliomas (LGGs), pediatric patients (<18 years), patients that followed other schemes of treatment outside the Stupp scheme and those who died for other reasons than GBM.

The OS, defined as the time from surgery to the date of death, and the progression-free survival (PFS), defined as the time from the first radio-chemotherapy treatment to the date of clinical or radiological progression according with the RANO criteria, were analyzed. Regarding the type of surgery, three groups were defined according with the postoperative MRI performed in the first 72 h: GTR (with no contrast-enhancing residual tissue visible on T1-injected MRI sequences), incomplete STR (with evidence of contrast-enhancing residual tumor) and Biopsy.

### 2.1. Histological and Molecular Analysis

The pathological evaluation had been performed at the Institute of Pathology, EOC, in Locarno (Switzerland) by experienced pathologists. MGMT promoter methylation evaluation, MGMT immunohistochemistry (IHC) and miRNA analysis have been done for each sample.

### 2.2. MGMT Promoter Methylation

Genomic DNA for MGMT analyses were obtained from three 8 μm thick formalin-fixed, paraffin-embedded (FFPE) tumor sections using automatic extraction (Maxwell, Promega, Madison, WI, USA). About 100 ng of DNA were subjected to bisulfite treatment using EZ DNA Methylation-GoldTM kit (Zymo Research, Irvine, CA, USA). Afterwards, the methylation status was assessed by PCR-pyrosequencing using MGMTPlus kit according to the recommended protocol (Diatech Pharmacogenetics, Jesi, Italy). This assay evaluates the methylation of six consecutive cytosines of MGMT promoter (chr10:131,265,507–131,265,556). The presence of methylation was determined applying a cut-off of 10%. This value was determined by calculating the limit of negative controls (DNA samples from 15 FFPE healthy brain tissues) for each cytosine (mean of methylation ratio adding 2× the Standard Deviation) assuming a Gaussian distribution of the raw signal from negative samples. 

### 2.3. MGMT Immunohistochemistry 

Three 1–2 μm thick sections obtained from whole FFPE tissue were deparaffinated, rehydrated and pretreated with citrate buffer pH6 in microwave oven for 20 min. Then the sections were treated overnight with the primary antibody anti-MGMT, clone MT3.1 (Chemicon International, Temecula, CA, USA) diluted 1/400 and followed by a polymeric detection system (Ultravision DAB Detection System, LabVision, Fremont, CA, USA) according to the manufacturer’s protocol. According to the literature, MGMT IHC positivity was scored when more than 5% of neoplastic cells showed intense nuclear staining [18,19]. Two pathologists scored the IHC independently.

### 2.4. miRNA Evaluation

The miRNA extraction was made from three 10 μm formalin-fixed, paraffin-embedded (FFPE) tumor sections using RecoverAll™ Total Nucleic Acid Isolation Kit for FFPE according to the manufacturer’s instructions. (ThermoFisher Scientific, Waltham, MA, USA). miRNA-specific retrotranscription was performed using TaqMan^®^ MicroRNA Reverse Transcription Kit and 5X primers included in inventoried TaqMan MicroRNA assays (Life Technologies, Carlsbad, CA, USA) for miR-21, miR-195, miR-767-3p, miR-196b, miR-648, miR-181d, miR-181c and RNU6B (used for endogenous control). Each sample was analyzed in three replicates using Universal Master Mix and assays from TaqMan MicroRNA assays (Life Technologies) as recommended by the manufacturer. As calibrators, we selected 12 normal brain samples from patients with cerebral arteriovenous malformations. Relative miRNA expression was calculated with the DDCt method.

### 2.5. Statistical Analyses

At first, mean and median values were calculated to summarize the results of each variable. The relative chi-square, CHI^2^_rel_, describes the statistical association existing among pairs of variables:
$$x_{rel}^{2}=\frac{1}{x_{max}^{2}}\underset{i}{\sum }\underset{j}{\sum }\frac{{\left(n_{i,j}-{\hat{n}}_{i,j}\right)}^{2}\;}{{\hat{n}}_{i,j}}$$
where n_i,j_ indicates the number of patients observed in row i and column j of the contingency table, hat-n is the expected value of counts given the statistical independence of the two considered variables, and with
$$x_{max}^{2}=N\;\cdot \;\min\left(m_{R}-1,\;m_{C}-1\right)$$
the maximum value taken by the chi-square statistic when the sample size is N and the contingency tables has *m_R_* rows and *m_C_* columns. The relative chi-square takes values in *[0, 1]*, with *0* indicating the lack of statistical association. 

OS and PFS curves for censored data were obtained using the Kaplan–Meier estimator.
$$\hat{S}\;\left(t\right)=\prod_{i:t_{i}\leq t}\left(1-\frac{d_{i}}{n_{i}}\right)$$
which is the estimate of the probability that life is longer than *t;* furthermore, *t_i_* is a time when at least one event (death) was observed, *d_i_ is* the number of events (deaths) that happened at time *t_i_*, and *n_i_* represents the individuals known to have survived up to time *t_i_*. Comparisons of curves given different molecular characterizations were performed by logrank tests. PFS curves were also estimated and tested within strata defined by the variable of surgery. 

Log-rank tests were performed on hypotheses of no difference among survival curves of groups defined by an explanatory variable [20].

All the analyses, graphs and reports were performed using the R software and the following R packages: bootstrap, survival.

For miRNA expression, there are no published cut-offs validated for GBM (nor in other diseases, nor for real-time experiments in general). Therefore, three different cut-offs for the evaluation of positive cases have been applied on the basis of similar studies published in the literature: Cut-off > 3; Cut-off > 1; Cut-off > median value. We present here only results given cut-off *> 3*, which represents the stronger methods for evaluating miRNA expression and which is the cutoff that we have already published in a previous paper [21]. In contrast, with a *cut-off* > 1 only a slight deviation is considered clinically relevant, while with a cut-off equal to the median values, the results could be too cohort dependent.

## 3. Results

### 3.1. General Consideration

In the ten year period (from January 2004 to December 2013), a total of 112 GBM IDH1 wt WHO grade IV patients were recruited from the centers involved in the study. Out of the evaluable cases, 39/108 (36.1%) showed a methylation of MGMT, whereas the remaining 69/108 (63.9%) were unmethylated. In addition, out of the evaluable cases, 54/98 (55.1%) of patients were positive for MGMT IHC and 44/98 (44.9%) were negative (Figure 1). 

From the neurosurgical point of view, a GTR has been achieved in 79 patients (70.5%), an STR was reached in 17 patients (15.2%), while in the remaining 16 cases (14.3%) the attending neurosurgeons opted for a biopsy (Table 1). All the clinicopathological and molecular data are summarized in Table 1.

### 3.2. miRNA Expression

As mentioned before, we report here the data obtained using a cut-off value >3, but the results with the two other values were superimposable (Table 2). In general, we observed strong miRNA overexpression (values > 3) for miR-21 and miR-196b, while downregulation (values < 0.33) was essentially observed for miR-767.3. The four remaining miRNAs showed normal expression (values between 0.333 and 3) in the majority of cases.

A multivariate analysis was performed between the level of miRNA expression and the positivity of MGMT IHC. A statistically significant correlation has been found between a positive MGMT IHC and the low expression of the following miRNAs: miR-181c (*p* = 0.001), miR-195 (*p* = 0.003), miR-648 (*p* = 0.03) and miR-767.3p (*p* < 0.001) (Table 3). 

The general expression pattern, when matched to MGMT promoter methylation status, showed interesting correlations. Indeed, the multivariate analysis performed between the level of miRNA expression and MGMT methylation found a significant association between unmethylated cases and the low expression of miR-181d (*p* = 0.02) and miR-648 (*p* = 0.004) (Table 4), while miR-196b high expression seems to be associated with methylated cases (*p* = 0.006) (Table 4). The four remaining miRNAs did not show any statistical association with MGMT promoter hypermethylation.

### 3.3. Overall Survival

The OS has been considered for both groups of patients included in the study (methylated vs. unmethylated GBM). As summarized in Figure 2A, the OS for methylated patients is significantly better (*p* = 0.006). The same statistical consideration has been performed for MGMT IHC (Figure 2B). The results show that a negative MGMT IHC is significantly correlated to a longer OS (*p* = 0.01).

Then we calculated the association of miRNA expression and OS. A significant correlation was found for miR-21 (*p* = 0.006) (Figure 3A), miR-195 (*p* = 0.02) (Figure 3B), miR-196b (*p* = 0.008) (Figure 3C) and miR-648 (*p* = 0.02) (Figure 3D).

### 3.4. Progression-Free Survival

Data have been analyzed also to define any significant relation with the PFS. A statistical correlation has been found between MGMT methylation status and PFS (*p* = 0.03) (Figure 4A) but also between GTR and PFS (*p* = 0.04) (Figure 4B). Contrary to the correlation with OS, we found no correlation between PFS and MGMT IHC, in addition, miRNA expression showed a trend, but never reached a statistically significant correlation with PFS.

## 4. Discussion

The possibility to predict the efficacy of therapeutic options today available for the management of GBMs, mainly chemotherapy and radiotherapy after surgery, is of primary relevance for clinicians and researchers. 

The last decade of research proved that specific molecular alterations may help in the early identification of patients who can be assigned to specific chemotherapies. The analysis of MGMT promoter hypermethylation (an alteration occurring in about 40% of GBMs) permits identifying GBM patients who can benefit from TMZ administration [3,6,10,22,23]. Indeed, it has been shown that the presence of MGMT promoter hypermethylation is a positive prognostic factor in GBMs treated with TMZ [3,6,10,22,23]. However, it has been reported that a group of non-methylated tumors may still be addressed, with good success, to the treatment with TMZ and, consequently, the identification of this subgroup of patients is of particular interest [23]. 

The present research proposal is inscribed in this context. Recent data pointed out that small molecules of RNA (named miRNA) can globally downregulate the expression of MGMT protein (not by a direct interaction but through one or several intermediate molecules [24], reaching a result comparable to the presence of promoter hypermethylation [11,25,26]. Furthermore, it has been demonstrated that the presence of these miRNAs may be associated with a good response to TMZ treatment [11]. At the moment, in the literature, these works have usually investigated a single miRNA in a single cohort of patients or in vitro experiment, with possible bias on the basis of the different cohorts. The information on this topic is very fragmented and preliminary. Consequently, our aim was to confirm these very promising and relevant data by analyzing several miRNA molecules in a cohort of patients with confirmed GBMs and treated with chemo radiotherapy after surgery according to the Stupp Protocol [3]. 

The present research analyzes seven different miRNAs and their possible implication in the modulation of MGMT.

With regard to the miRNAs, we observed that not all the miRNAs, selected on the basis of the clinical relevance reported in the literature, have similar trends. Two miRNAs (miR-21 and miR-196b) are generally overexpressed in GBMs, while one is usually downregulated (miR-767.3) and the four remaining do not have a particular pattern of expression.

The first interesting result is the significant correlation between the low expression of four out of seven miRNAs (miR-181c, miR-195, miR-648 and miR-767.3p) and the positivity for MGMT expression evaluated through IHC. These data correlate with the ones in the literature reporting that these miRNAs are downregulated in patients not responding to TMZ, which are generally the ones with high MGMT expression by IHC [27]. In addition, these miRNAs, in general, inhibit MGMT expression so it is reasonable that their low expression is associated with normal MGMT expression by IHC [15].

The second important result is the confirmation of the above-cited correlation looking at the methylation of the MGMT. In fact, analyzing miR648, the only one that has a statistically significant association with both MGMT IHC and MGMT methylation, we found that it is associated with the absence of MGMT methylation. The correlation of the expression of this miRNA and MGMT methylation has been reported in the literature, and logically confirms the data aforementioned for IHC because MGMT IHC and methylation are opposite situations [14,15]. Indeed, both low expression of miRNAs and the absence of MGMT methylation should bring about an expression of MGMT protein, and consequently, a positivity for MGMT protein. All the other miRNAs that present a statistically significant association between their low expression and positive MGMT IHC (i.e., miR181c, miR195 and miR767.3) do not present a significant association with MGMT methylation. This discrepancy could be due to the fact that the miRNAs analyzed in this study do not interact directly with MGMT but regulate MGMT indirectly through other mediators; this could result in different behavior towards MGMT. The same explanation can justify the fact that miR181d and miR196b expression associate with MGMT methylation (miR181d low expression with unmethylated cases and miR196B with methylated cases) but not with MGMT IHC.

Salient results have been found in terms of OS and PFS. At first, we confirmed the positive correlation of better OS and PFS (*p* = 0.006 and *p* = 0.03, respectively) and MGMT promoter hypermethylation, a finding indicating that our cohort is fully representative. Furthermore, the study also takes into account the different outcome in relation to the type of surgery performed. In consonance with our previous published work and the evidence in the literature, patients undergoing a GTR show a more favorable clinical outcome in terms of PFS (*p* = 0.04) [28]. The better clinical outcome is also noted in patients with negative IHC (OS *p* = 0.01), thus confirming the reliability of the analysis and the inverse correlation between negative IHC and presence of methylation. 

Finally, and notably, the present work tried to find a significant relation between miRNA expression and OS. Indeed, in four out of seven of the miRNAs analyzed, this relation has results that were statistically discernible (miR-21, miR-195, miR-196b and miR-648 with *p* = 0.006, *p* = 0.02, *p* = 0.008 and *p* = 0.02, respectively), confirming that miRNAs play a pivotal role in the modulation of the MGMT and, consequently, in the clinical outcome of patients with GBM.

## 5. Conclusions

The present study represents, to the best of our knowledge, one of the first scientific attempts to investigate the expression of several miRNAs in the same cohort. As there are no firmly established methods for the assessment of miRNA expression in cancer (not only in GBM), it is possible to assume a bias when different miRNAs are evaluated in different cohorts. We believe that the real clinical relevance of miRNA expression may emerge in studies similar to the present one, when multiple miRNAs are evaluated simultaneously. The same can be hypothesized for the assessment of MGMT promoter methylation analysis. Indeed, the MGMT evaluation method applied in our study (pyrosequencing) is reliable, robust and applied in routine diagnosis as well. However, other methods are available on the market for the evaluation of such a marker, and pyrosequencing, compared to other methodologies such as the quantitative MGMT methylation-specific PCR (qMSP) MGMT test, has a limitation; it does not permit differentiation between the true unmethylated form from the MGMT “grey zone”. This could bring to an imprecise evaluation of the response to therapies because the grey zone, defined by specific cut-offs by Hegi and colleagues, has a behavior more similar to the methylated cases than the unmethylated ones [29]. However, we must emphasize that all methods for assessing the methylation status of the MGMT promoter are valid and can be used confidently.

Returning to miRNA evaluation, our data are superimposable when different cut-offs were used to assess miRNA expression, thus strengthening our conclusions. Our data reinforce the notion of the clinical relevance of miRNA expression as an alternative method to assess promoter methylation in the regulation of MGMT expression. If confirmed in other cohorts, possibly including a larger series of GBMs, our data open the door to a future diagnostic role of miRNA expression in predicting the efficacy of chemoradiation in GBM. 

## Figures and Tables

**Figure 1 jcm-12-02061-f001:**
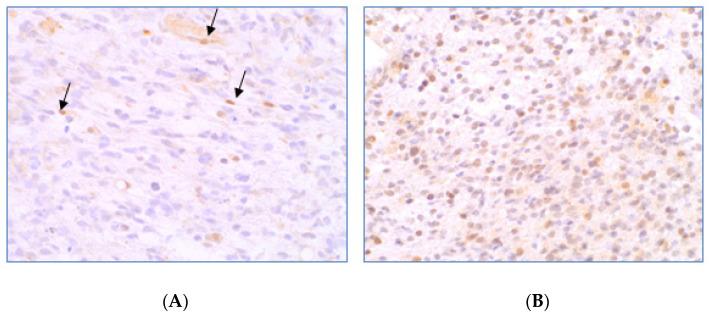
MGMT expression evaluated by IHC. (**A**) An example of a GBM with negative staining of MGMT protein. The arrows indicate internal positive control, namely inflammatory cells and endothelial cells. (**B**) An example of a GBM showing expression of MGMT protein.

**Figure 2 jcm-12-02061-f002:**
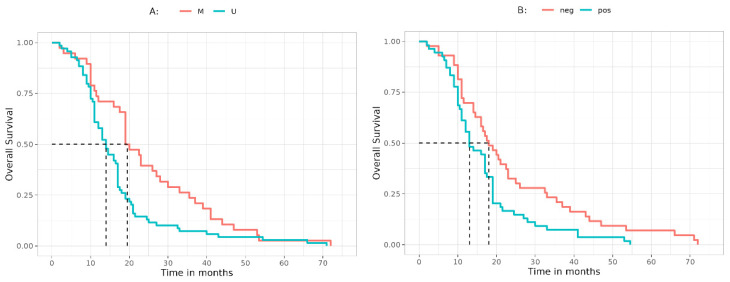
OS curves representing the survival of patients (in terms of months) on the basis of MGMT. (**A**) OS evaluated on MGMT methylation status. Dashed vertical lines: median survival times are 19.5 months for M and 14.0 for U. (**B**) OS evaluated on MGMT expression by IHC. Dashed vertical lines: median survival times are 18 months for neg and 13 months for pos groups. Abbreviations: IHC, immunohistochemistry; M, methylated; neg, negative; OS, overall survival; pos, positive; U, unmethylated.

**Figure 3 jcm-12-02061-f003:**
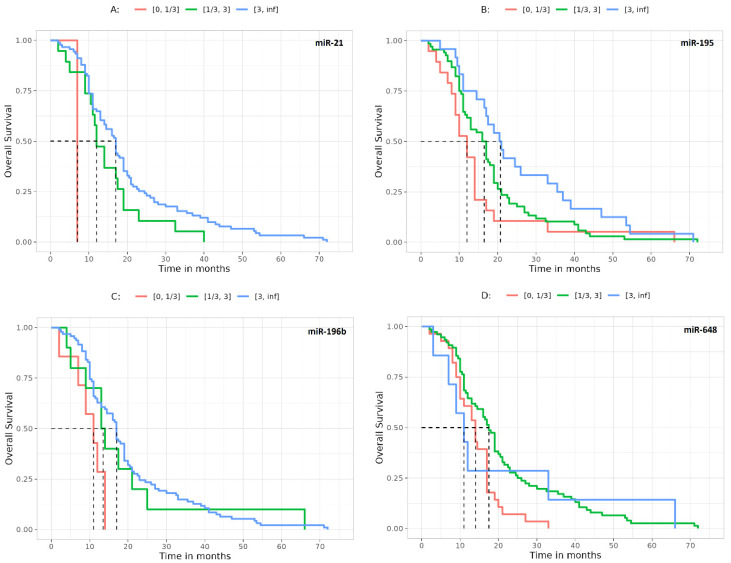
OS survival curves representing the survival of patients (in terms of months) on the basis of different miRNA expression. Only the curves of the miRNAs associated to a significant result are reported. (**A**) OS evaluated on miR-21. Dashed vertical lines: median survival times (months) [0, 1/3]: 7, [1/3, 3]: 12, [3, inf]: 17. (**B**) OS evaluated on miR-195. Dashed vertical lines: median survival times (months) [0, 1/3]: 12, [1/3, 3]: 16.5, [3, inf]: 20.8. (**C**) OS evaluated on miR-196b expression. Dashed vertical lines: median survival times (months) [0, 1/3]: 11, [1/3, 3]: 13.5, [3, inf]: 17. (**D**) OS evaluated on miR-648. Dashed vertical lines: median survival times (months) [0, 1/3]: 14, [1/3, 3]: 17.5, [3, inf]: 11. Abbreviations: OS, overall survival.

**Figure 4 jcm-12-02061-f004:**
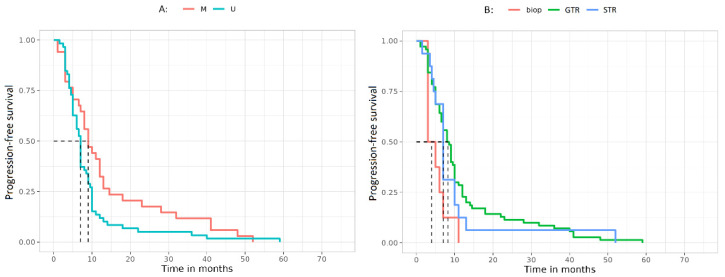
(**A**) PFS curves representing the survival of patients (in terms of months) on the basis of MGMT methylation status. Dashed vertical lines: median survival times (months) U: 7, M: 9. (**B**) PFS curves representing the survival of patients (in terms of months) on the basis of surgery type. Dashed vertical lines: median survival times (months) biop: 4, GTR: 8.25, STR: 7. Abbreviations: biop, biopsy; GTR, Gross Total Resection; M, methylated; PFS, progression-free survival; STR, Subtotal Tumor Resection; U, unmethylated.

**Table 1 jcm-12-02061-t001:** Patients’ characteristics.

Patient’s Characteristics	
**Age**	
28–85 years	
	**no. (%)**
**Sex**	
Male	57/112 (50.9%)
Female	55/112 (49.1%)
**Stage of disease at diagnosis**	
IV	112/112 (100%)
**Histologic Type**	
GBM IDH1 wt WHO grade IV	112/112 (100%)
**MGMT promoter methylation**	
M	39/108 (36.1%)
UM	69/108 (63.9%)
**MGMT IHC**	
pos	54/98 (55.1%)
neg	44/98 (44.9%)
**Neurosurgery**	
GTR	79/112 (70.5%)
STR	17/112 (15.2%)
BIOPSY	16/112 (14.3%)

Patients’ clinicopathological characteristics and molecular data. Abbreviations: GBM, glioblastoma multiforme; GTR, gross total resection; IHC, immunohistochemistry; M, methylated; neg, negative; pos, positive; STR, subtotal resection; UM, unmethylated.

**Table 2 jcm-12-02061-t002:** miRNA distribution.

miRNA	miRNA Expression Values		
	<0.333	0.333–3	>3
**miR-181c**	39/112 (34.8%)	62/112 (55.4%)	11/112 (9.8%)
**miR-181d**	38/112 (33.9%)	71/112 (63.4%)	3/112 (2.7%)
**miR-21**	1/112 (0.9%)	20/112 (17.9%)	91/112 (81.2%)
**miR-195**	19/112 (17%)	69/112 (61.6%)	24/112 (21.4%)
**miR-196b**	7/112 (6.2%)	10/112 (9%)	95/112 (84.8%)
**miR-648**	28/112 (25%)	77/112 (68.7%)	7/112 (6.3%)
**miR-767.3p**	47/86 (54.6%)	17/86 (19.8%)	22/86 (25.6%)

miRNA expression distribution considering as threshold the value 3. Abbreviations: miRNA, microRNA.

**Table 3 jcm-12-02061-t003:** MGMT IHC and miRNA bivariate analysis.

		IHC MGMT		*p*
		neg	pos	
**miR-181c**	**<0.333**	10/98 (10.2%)	22/98 (22.5%)	**0.0015**
	**0.333–3**	24/98 (24.5%)	31/98 (31.6%)	
	**>3**	10/98 (10.2%)	1/98 (1%)	
**miR-181d**	**<0.333**	11/98 (11.2%)	22/98 (22.4%)	0.0560
	**0.333–3**	30/98 (30.6%)	32/98 (32.7%)	
	**>3**	3/98 (3.1%)	0/98 (0%)	
**miR-21**	**<0.333**	0/98 (0%)	1/98 (1%)	0.7846
	**0.333–3**	7/98 (7.1%)	11/98 (11.2%)	
	**>3**	37/98 (37.8%)	42/98 (42.9%)	
**miR-195**	**<0.333**	6/98 (6.1%)	10/98 (10.2%)	**0.0025**
	**0.333–3**	21/98 (21.4%)	39/98 (39.8%)	
	**>3**	17/98 (17.4%)	5/98 (5.1%)	
**miR-196b**	**<0.333**	2/98 (2%)	4/98 (4.1%)	0.8216
	**0.333–3**	4/98 (4.1%)	4/98 (4.1%)	
	**>3**	38/98 (38.8%)	46/98 (46.9%)	
**miR-648**	**<0.333**	4/98 (4.1%)	17/98 (17.3%)	**0.0230**
	**0.333–3**	37/98 (37.8%)	35/98 (35.7%)	
	**>3**	3/98 (3.1%)	2/98 (2%)	
**miR-767.3p**	**<0.333**	12/98 (15.8%)	28/98 (36.8%)	**0.0005**
	**0.333–3**	11/98 (14.5%)	5/98 (6.6%)	
	**>3**	15/98 (19.7%)	5/98 (6.6%)	

Relative frequencies and *p* values obtained from bivariate analysis between miRNA expression, based on the cut-off value > 3 and MGMT IHC results. Level of significance: *p* < 0.05 (in bold). Abbreviations: IHC, immunohistochemistry; neg, negative; *p*, *p* value; pos, positive.

**Table 4 jcm-12-02061-t004:** MGMT methylation and miRNA bivariate analysis.

		Met MGMT		*p*
		M	U	
**miR-181c**	**<0.333**	10/108 (9.3%)	26/108 (24.1%)	0.4288
	**0.333–3**	24/108 (22.2%)	37/108 (34.3%)	
	**>3**	5/108 (4.6%)	6/108 (5.5%)	
**miR-181d**	**<0.333**	9/108 (8.3%)	26/108 (24.1%)	**0.0245**
	**0.333–3**	27/108 (25%)	43/108 (39.8%)	
	**>3**	3/108 (2.8%)	0/108 (0%)	
**miR-21**	**<0.333**	0/108 (0%)	0/108 (0%)	1.0000
	**0.333–3**	6/108 (5.6%)	12/108 (11.1%)	
	**>3**	33/108 (30.6)	57/108 (52.8%)	
**miR-195**	**<0.333**	3/108 (2.8%)	13/108 (12%)	0.2984
	**0.333–3**	26/108 (24.1%)	42/108 (38.9%)	
	**>3**	10/108 (9.3%)	14/108 (12.9%)	
**miR-196b**	**<0.333**	0/108 (0%)	4/108 (3.7%)	**0.0060**
	**0.333–3**	0/108 (0%)	10/108 (9.3%)	
	**>3**	39/108 (36.1%)	55/108 (50.9%)	
**miR-648**	**<0.333**	3/108 (2.8%)	23/108 (21.3%)	**0.0035**
	**0.333–3**	34/108 (31.5%)	41/108 (37.9%)	
	**>3**	2/108 (1.9%)	5/108 (4.6%)	
**miR-767.3p**	**<0.333**	14/83 (16.9%)	31/83 (37.3%)	0.8591
	**0.333–3**	6/83 (7.2%)	10/83 (12.1%)	
	**>3**	8/83 (9.6%)	14/83 (16.9%)	

Relative frequencies and *p* values obtained from bivariate analysis between miRNA expression based on the cut-off > 3 and MGMT methylation results. Level of significance: *p* < 0.05 (in bold). Abbreviations: M, methylated; Met, methylation; *p*, *p* value; U, unmethylated.

## Data Availability

The datasets used and analyzed during the current study are available from the corresponding author on reasonable request. The data are not publicly available due to institutional policy.

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
