# Peer review of "Identification of MGMT Downregulation Induced by miRNA in Glioblastoma and Possible Effect on Temozolomide Sensitivity"

_jcm, 2023, doi:10.3390/jcm12052061_

Round 1

Reviewer 1 Report

The authors assess the association between MGMT methylation and expression with miRNA expression and the response to Temozolomide in GBM and overall patient outcome.    

MGMT promoter methylation is classified as (un)methylated. The authors performed pyrosequencing to determine methylation status.    

However, as array based, or PCR based determination of methylation status is also widely used, it would be important to at least discuss (even better: provide) data from these alternative methods and their respective cutoffs (see e.g. PMID30514777).

Please provide references for the miRNA cutoff justification. Did an evaluation as continuous variable confirm the results?    

Were IHCs scored by a single rater? Ideally, two independent raters would assess the data. Did the authors look add additional parameters (e.g. differences in (sub)cellular distribution of MGMT?).           

Axis labels are missing (e.g. Fig 2). What kind of p-values are reported? If Cox-PH models were used, please check and comment on the proportionalhazards assumption, especially for Figure 3D.    

What is the correation between the different miRs, especially the ones linked to outcome?    

Is there an added benefit of MGMT IHC, miR expression? Please provide multivariate survival data, and test for interactions.      

Minor:    

p-values do not need to be reported with 8 digits (Tbl.3)    

Please provide median survival / PFS data for better comparability with data from literature.    

Please thoroughly check the naming of types of analyses (e.g. multivariate vs bivariate ... )    

A schematic visualizing the proposed model of regulation and associations between MGMT methylation, expression and miR abundance would help.    

Author Response

Dear reviewer,

Reviewer1

Comments and Suggestions for Authors

At first, we would like to thank the reviewer 1 for his/her very helpful and specific comments and suggestions that will help to improve the manuscript. Please see below, in red, the detailed reply to all the points raised by the reviewer. The number of pages and lines reported refer to the text with track changes not yet accepted.

The authors assess the association between MGMT methylation and expression with miRNA expression and the response to Temozolomide in GBM and overall patient outcome.

1.MGMT promoter methylation is classified as (un)methylated. The authors performed pyrosequencing to determine methylation status. However, as array based, or PCR based determination of methylation status is also widely used, it would be important to at least discuss (even better: provide) data from these alternative methods and their respective cutoffs (see e.g. PMID30514777).

To date, for the assessment of MGMT promoter methylation there are multiple techniques, none of which is firmly established or suggested by international guidelines. Therefore we used the method that we know better and apply at diagnostic level in our institution for our patients. However, taking inspiration from reviewer comment, we decided to include the discussion of the topic in the conclusions section (page 14, lines 378-391, reference 28).

  1. Please provide references for the miRNA cutoff justification. Did an evaluation as continuous variable confirm the results?

As reported in the manuscript (page 4, lines 181-189), for the evaluation of miRNA expression (as for all the other markers which are evaluated in a quantitative or at least semi-quantitative manner by a real-time PCR approach) there are no published cut-offs, validated and widely recognized by the scientific community or by international guidelines. According to our previous experience (Forcella et al, Cancer Biomarkers 2018, 21: 591-601), we choose the Cut-off equal to 3 between the three feasible different cut-offs for real-time PCR experiments (Cut-off > 3; Cut-off > 1; Cut-off > median value) because we believe it represents the stronger value for evaluating miRNA expression. Therefore, we suggest as reference our previous study (reference 20).

  1. Were IHCs scored by a single rater? Ideally, two independent raters would assess the data. Did the authors look add additional parameters (e.g. differences in (sub)cellular distribution of MGMT?)

The IHCs were scored indipendently by pathologists with experience in the field and who do the evaluation even for the normal diagnostic routine; no additional MGMT parameters has been evaluated. We would like to thank the reviewer for this comment and we have modified the manuscript accordingly (page 3, line 138).

  1. Axis labels are missing (e.g. Fig 2).

We have modified the figures introducing axis labels, in addition also median survival times are shown (Figures 2A+B, 3A+B+C+D, 4A+B).

What kind of p-values are reported? If Cox-PH models were used, please check and comment on the proportional hazards assumption, especially for Figure 3D.    

The Figure 3D “OS evaluated on miR-648" shows curves where risk is likely to change over time (months), especially for group (0.333,3] which differ at late times with respect to the other two groups. Two survival curves crossed thus we have evidence that proportional hazard assumption could be violated.

Nevertheless, we used a statistical procedure that implements the G-rho family of Harrington and Fleming (1982), with weights on each death of S(t), where S(t) is the Kaplan-Meier estimate of survival. With rho = 0 this is the log-rank or Mantel-Haenszel test, as we did. The p-value refers to a Chi Square test comparing observed and expected number of events under the null hypothesis stating that no difference is present among survival functions.

Please see the textbook-length reference “Survival analysis with interval-censored data, section 2.2 pag. 41-45, by Kris Bogaerts, Arnost Komarek, Emmanuel Lesaffre, and the following papers:

Harrington, D. P. and Fleming, T. R. (1982). A class of rank test procedures for censored survival data. Biometrika, 553-566.

Peto R. Peto and Peto, J. (1972) Asymptotically efficient rank invariant test procedures (with discussion), JRSSA, 185-206.

Prentice, R. and Marek, P. (1979) A qualitative discrepancy between censored data rank tests, Biometrics, 861–867.

We have modified Figure 3 and we have modified the text (page 4, lines 176-177, reference 19).

  1. What is the correlation between the different miRs, especially the ones linked to outcome? Is there an added benefit of MGMT IHC, miR expression? Please provide multivariate survival data, and test for interactions.   

The aim of the present study was the clinical relationship between the expression of several miRNA and clinical data. We intended the evaluation of MGMT (promoter methylation and protein expression) as an indication of the representativity of our cohort. The comparison of the relative expression of miRNA and the relationship with MGMT alterations is the aim of a further study that is now under preparation. As a consequence, we have not added any new data.

As regards the comment on the multivariate analysis, we have not produced multivariate survival data because with the size cohort of our manuscript it is not possible to obtain reliable information from this analysis.

  1. Minor:    

p-values do not need to be reported with 8 digits (Tbl.3)   

We changed the text accordingly (Table 3 and 4).

Please provide median survival / PFS data for better comparability with data from literature.    

We changed the text accordingly (please see the paragraphs added in the captions of Figures 2, 3 and 4).

Please thoroughly check the naming of types of analyses (e.g. multivariate vs bivariate ... )    

We adhered to the common convention where the analysis of just one response variable is called univariate, even if several other explanatory variables are considered in the same model.

A schematic visualizing the proposed model of regulation and associations between MGMT methylation, expression and miR abundance would help.    

The interaction between MGMT and miRNAs is indirect and mediated by other molecules on which the miRNAs described in our paper act. In addition, these miRNAs have a vast number of ligands (see http://www.mirdb.org/ for details) that cannot be represented in a schematic visualizing due to their large number. For these reasons we have not added a scheme concerning this topic, which will be further explored in the paper in preparation. However, we have add this information in the text (see rows 306-307)

Reviewer 2 Report

In this study Cardia et al., the authors investigated whether a pattern of miRNA expression correlates with response to the treatment with temolozomide (TMZ) and his clinical efficacy.

The manuscript contains some new information. Abstract is clear and accurately describe the content of the article. Experiments are describe comprehensively and the interpretations and conclusions are justified by the results. Although the priority for publishing of this article is not the highest but final recommendation is: accept after minor correction.

General

The major concern with this study that the authors introduce their subject with the old-World Health Organization (WHO 2016) classification of tumors of the central nervous system (CNS). Authors should update it with the new one (Louis DN et al., Neuro Oncol. 2021 – 10.1093/neuonc/noab106). The 2021 classification largely takes into account the advent of new technologies and there are changes in nomenclature and taxonomic category of CNS tumors.

The second issue concern choosing of miRNAs. It is not well explained why these (7) and no other miRNAs are being studied. Provide rational explanation.

The third issue concern molecular mechanism. The authors did not provide any direct evidences that the tested miRNAs directly regulate the expression of MGMT (any strong evidence to support this hypothesis).

Does MGMT 3′ UTR contain binding sites for the tested miRNAs? (prediction programs such as DIANA-MicroT, TargetScan, miRranda and PicTar  should be useful). Do the tested miRNAs effect MGMT expression by direct interaction with the MGMT 3’ UTR? Do the tested miRNAs sensitize cells to TMZ treatment?

Minor points:

- PR (Partial Resection) or STR (subtotal resection) - it should be unified

- Table 1 - provide the data concerning age of the patient cohort

- what is the status of IDH in the tested samples?

- lack of dots at the end of some sentences (lines 77, 144) and extra dot – line 157

- line 217 - incorrect p value was given, it should be  - miR-648 (p=0.02)

- line226 - are you sure that correct conclusions were drawn after the multivariable analysis performed between the level of miR-196b and MGMT methylation? “…while miR-196b low expression seems to be associated with methylated cases (p=0.006) (Table 4).” It should be rather “…while miR-196b high expression……….”

- line 303 -309 - in discussion – “In fact, the low expression of miRNA miR-181d and miR-648 are statistically associated with the absence of MGMT methylation. ……..Indeed both low expression of miRNAs and the absence of MGMT methylation bring to an expression of MGMT protein and consequently a positivity for MGMT protein”.

It is true that statistical analyses demonstrate a significant association between low expression of miRNA miR-181d and unmethylated cases but statistically significant correlation has not been found between a positive MGMT IHC and level of miRNA miR-181d expression. That correlation was observed between miRNA miR-181c. Please be more precise.

 I hope that these changes have made the manuscript more appropriate for publication.

Author Response

Dear reviewer 2,

Reviewer2

Comments and Suggestions for Authors

We would like to thank the second reviewer for the relevant and interesting comments that will help to improve our paper. Please see below, in red, our detailed point by point reply. The number of pages and lines reported refer to the text with track changes not yet accepted.

In this study Cardia et al., the authors investigated whether a pattern of miRNA expression correlates with response to the treatment with temolozomide (TMZ) and his clinical efficacy.

The manuscript contains some new information. Abstract is clear and accurately describe the content of the article. Experiments are describe comprehensively and the interpretations and conclusions are justified by the results. Although the priority for publishing of this article is not the highest but final recommendation is: accept after minor correction.

General

1.The major concern with this study that the authors introduce their subject with the old-World Health Organization (WHO 2016) classification of tumors of the central nervous system (CNS). Authors should update it with the new one (Louis DN et al., Neuro Oncol. 2021 – 10.1093/neuonc/noab106). The 2021 classification largely takes into account the advent of new technologies and there are changes in nomenclature and taxonomic category of CNS tumors.

Thank you for this relevant specification. We have changed the text applying the 2021 classification (page 2, line 97; page5, line 193; Table1). All the cases are GBM IDH1 wild-type, WHO grade IV.

  1. The second issue concern choosing of miRNAs. It is not well explained why these (7) and no other miRNAs are being studied. Provide rational explanation.

The miRNA that we analyzed are the ones that, at the time of the start of the project, seemed to be the most relevant in glioblastomas at clinical level. In the revised version of the manuscript, we have introduced some references reporting the relevance of these miRNAs (page 2, lines 72-74). The list of references have been changed accordingly (references 14-16).

  1. The third issue concern molecular mechanism. The authors did not provide any direct evidences that the tested miRNAs directly regulate the expression of MGMT (any strong evidence to support this hypothesis).

Does MGMT 3′ UTR contain binding sites for the tested miRNAs? (prediction programs such as DIANA-MicroT, TargetScan, miRranda and PicTar  should be useful). Do the tested miRNAs effect MGMT expression by direct interaction with the MGMT 3’ UTR? Do the tested miRNAs sensitize cells to TMZ treatment?

We have not elaborated on this point by applying precise prediction programs because the mechanism of action of these miRNAs has already been reported in the literature. In addition we decided to test these miRNAs for their clinical relevance reported in literature and not for their hypothetic binding to MGMT. Following your suggestion, we have gone into more detail on their function by introducing a small paragraph in the introduction (page 2, lines 68-71, references 11-13). We hope that our changes will make this part clearer.

  1. Minor points:

- PR (Partial Resection) or STR (subtotal resection) - it should be unified

Thank you for this clarification, we changed the text accordingly.

- Table 1 - provide the data concerning age of the patient cohort

We have added the age of the patients in Table 1.

- what is the status of IDH in the tested samples?

All the patients are IDH1 wild-type. We have reported this classification at page 2 line 97 and at page 5 line 193.

- lack of dots at the end of some sentences (lines 77, 144) and extra dot – line 157

We changed the text accordingly, thank you.

- line 217 - incorrect p value was given, it should be  - miR-648 (p=0.02)

We changed the text accordingly, thank you.

- line226 - are you sure that correct conclusions were drawn after the multivariable analysis performed between the level of miR-196b and MGMT methylation? “…while miR-196b low expression seems to be associated with methylated cases (p=0.006) (Table 4).” It should be rather “…while miR-196b high expression……….”

We agree with this conclusion and we changed the text accordingly, thank you (page 8, line 239).

- line 303 -309 - in discussion – “In fact, the low expression of miRNA miR-181d and miR-648 are statistically associated with the absence of MGMT methylation. ……..Indeed both low expression of miRNAs and the absence of MGMT methylation bring to an expression of MGMT protein and consequently a positivity for MGMT protein”.

It is true that statistical analyses demonstrate a significant association between low expression of miRNA miR-181d and unmethylated cases but statistically significant correlation has not been found between a positive MGMT IHC and level of miRNA miR-181d expression. That correlation was observed between miRNA miR-181c. Please be more precise.

Thank you for this precisation, we agree and we changed the discussion accordingly (page 13, lines 338-346).

I hope that these changes have made the manuscript more appropriate for publication.
